# Cloning and Functional Analysis of a Zeaxanthin Epoxidase Gene in *Ulva prolifera*

**DOI:** 10.3390/biology13090695

**Published:** 2024-09-05

**Authors:** Hongyan He, Xiuwen Yang, Aurang Zeb, Jiasi Liu, Huiyue Gu, Jieru Yang, Wenyu Xiang, Songdong Shen

**Affiliations:** 1School of Biology & Basic Medical Sciences, Soochow University, Suzhou 215101, China; 2Suzhou Industrial Park Environmental Law Enforcement Brigade, Suzhou 215021, China; xwy@sipac.gov.cn

**Keywords:** *Ulva prolifera*, zeaxanthin epoxidase, high-salt stress, overexpression

## Abstract

**Simple Summary:**

The xanthophyll cycle plays an important role in plants’ responses to abiotic stress. Zeaxanthin epoxidase (*ZEP*) contributes to the xanthophyll cycle and the biosynthesis of abscisic acid (ABA) and carotenoids. The biological function of *ZEP* has been extensively studied in higher plants but has not been reported in *Ulva prolifera*. In this study, the relative content of xanthophylls was determined in *U. prolifera*. We also cloned and analyzed a *ZEP* gene from *U. prolifera*, namely *UpZEP*. The expression and biological function of the *UpZEP* gene were examined. The results of this study provide a theoretical basis for understanding the high-salt resistance of *U. prolifera*.

**Abstract:**

The xanthophyll cycle is a photoprotective mechanism in plants and algae, which protects the photosynthetic system from excess light damage under abiotic stress. Zeaxanthin is considered to play a pivotal role in this process. In this study, the relative content of xanthophylls was determined using HPLC-MS/MS in *Ulva prolifera* exposed to different salinities. The results showed that high-salt stress significantly increased the relative content of xanthophylls and led to the accumulation of zeaxanthin. It was speculated that the accumulated zeaxanthin may contribute to the response of *U. prolifera* to high-salt stress. Zeaxanthin epoxidase (*ZEP*) is a key enzyme in the xanthophyll cycle and is also involved in the synthesis of abscisic acid and carotenoids. In order to explore the biological function of *ZEP*, a *ZEP* gene was cloned and identified from *U. prolifera*. The CDS of *UpZEP* is 1122 bp and encodes 373 amino acids. Phylogenetic analysis showed that *UpZEP* clusters within a clade of green algae. The results of qRT-PCR showed that high-salt stress induced the expression of *UpZEP*. In addition, heterologous overexpression of the *UpZEP* gene in yeast and *Chlamydomonas reinhardtii* improved the salt tolerance of transgenic organisms. In conclusion, the *UpZEP* gene may be involved in the response of *U. prolifera* to high-salt stress and can improve the high-salt tolerance of transgenic organisms.

## 1. Introduction

Green tides are a marine ecological anomaly that usually occur in eutrophicated estuaries, inner bays and urban dense coastal zones [1]. Under certain environmental conditions, green seaweed rapidly grows and propagates, accumulating in piles and thus causing green tides [2]. The first large-scale outbreak of green tides occurred in France, followed by the United States, Japan and China [3]. In recent years, green tides have increasingly become a global environmental concern because of the rising severity, geographical scale and frequency of macroalgal blooms [4]. Since 2007, green tides have been breaking out along the Yellow Sea coast every year. Green tides cover a wide area and have a large influence, seriously threatening the ecological safety of China [3,5]. The common species causing green tide disasters in China are *Ulva*, *Cladophora* and *Chaetomorpha*, with *Ulva prolifera* being the predominant species [6,7].

*U*. *prolifera* grows in nearshore ponds, estuaries and intertidal zones, where the environment is very complex and susceptible to abiotic stresses, such as salt stress, light stress and temperature stress [8]. To cope with abiotic stress, *U*. *prolifera* has evolved numerous defense mechanisms [9,10]. Salt stress is one of the abiotic stresses that restricts the growth and development of *U*. *prolifera* [11,12]. In general, both too high and too low salinity have adverse effects on photosynthesis, respiration and other physiological activities of *U*. *prolifera* [11]. In order to understand how *U*. *prolifera* responds to salt stress, biochemical and physiological response mechanisms have been extensively investigated [12,13]. Under salt stress, the photosynthetic performance of *U. prolifera* decreases, and the expression levels of several stress-resistance protein-related genes, such as heat shock protein and tubulin, are significantly up-regulated. These play an important role in sustaining life activities [12]. *U. prolifera* can also cope with high-salt stress by enhancing non-photochemical quenching (NPQ) [13]. Moreover, *U. prolifera* cells accumulate carbohydrates, such as glucose and fructan, under salt stress, with their main functions being osmotic protection and scavenging reactive oxide species (ROS) [14]. Interestingly, *U. prolifera* has salinity-dependent morphological variations; it has more branches under low salinity conditions and longer branches under high salt conditions [15].

In general, plants capture and use light energy efficiently. However, under environmental stress, plants can absorb excess light energy, which can lead to the destruction of the photosystem [16]. To avoid destroying the photosystems, plants have evolved a variety of photoprotection strategies, among which the xanthophyll cycle is considered an important photoprotection mechanism [17]. The xanthophyll cycle involves three components: violaxanthin, antheraxanthin and zeaxanthin. When plants receive excess light energy, zeaxanthin is produced through two de-epoxidation reactions under the action of violaxanthin de-epoxidase (VDE). Conversely, when light energy is insufficient, violaxanthin is produced by two epoxidation reactions under the action of zeaxanthin epoxidase, with antheraxanthin serving as the intermediate product in both the de-epoxidation and epoxidation reactions [18]. The xanthophylls are uniformly distributed in the thylakoid membrane, where they bind to the light-harvesting complex (LHC), which changes its conformation, leading to the dissipation of excess light energy [19]. Zeaxanthin, a xanthophyll with strong antioxidant activity, is widely used in feed, cosmetics, food and medical treatment [20]. Moreover, zeaxanthin can eliminate ROS generated by photosynthesis, such as triplet oxygen, singlet oxygen and superoxide anion, thereby safeguarding the integrity of the photosystem [21].

Zeaxanthin epoxidase (*ZEP*) is a bi-functional monooxygenase that epoxidizes zeaxanthin to antheraxanthin and then further to violaxanthin. This epoxidation reaction not only contributes to the xanthophyll cycle but also to the biosynthesis of abscisic acid (ABA) and carotenoids [22,23]. Several reports have suggested that *ZEP* plays an important role in plant growth, development and abiotic stress resistance [24]. Under abiotic stress, plants can regulate the content of carotenoids and other secondary metabolites by regulating the expression of the *ZEP* gene, thereby removing excessive ROS caused by abiotic stress. The expression of *ZEP* in tobacco and tomato roots significantly increased under drought stress [25,26]. The expression of *ZEP* in *Arabidopsis thaliana* roots was up-regulated under drought stress, and the content of ABA in roots and leaves also increased [27]. In addition, overexpression of a *ZEP* gene from *Medicago sativa* in tobacco significantly improved its salt tolerance [28]. The high expression of *ZEP* genes involved in ABA synthesis in *Brassica napus* increased drought tolerance [29]. These studies demonstrate that higher plants respond to abiotic stress by up-regulating *ZEP* gene expression.

The biological function of *ZEP* has been extensively studied in higher plants but has not been reported in *U. prolifera*. Here, the HPLC-MS/MS technique was employed to determine the relative contents of lutein, violaxanthin, antheraxanthin and zeaxanthin in *U. prolifera* cultivated under normal and high-salt conditions. Then, a *ZEP* gene was cloned and analyzed from *U. prolifera*, and the expression of the *UpZEP* gene was analyzed using qRT-PCR. In addition, to detect the function of the *UpZEP* gene, it was heterologously overexpressed in *Chlamydomonas reinhardtii* and *Saccharomyces cerevisiae*, which provided scientific insight into the high-salt resistance of *U. prolifera*.

## 2. Materials and Methods

### 2.1. Alga Material and Treatment

*U. prolifera* was collected from the coast of Qingdao, China (36°48′39.75″ N; 121°38′10.88″ E). Pure lines were obtained through genetic and morphological identification and were cultured in our laboratory. These lines were then cultured in distilled seawater supplemented with Von Stosch’s Enriched (VSE) medium under conditions of 20 ± 1 °C, a light intensity of 120 µmol photons m^−2^·s^−1^ and a photoperiod of 12 L:12 D [30]. After 15 d of cultivation, the thalli with healthy growth and similar lengths were selected and placed in a constant temperature light incubator for high-salt stress (50‰) treatment at different time points, with other culture conditions remaining unchanged.

The *C. reinhardtii* wild-type (WT) strain was provided by Dr. Yingjuan Wang (College of Life Science, Northwest University, Xi’an, China) and was cultivated in Tris-acetate-phosphate (TAP) medium under the following conditions: a light intensity of 50–60 μmol photons m^−2^·s^−1^, a photoperiod of 12 L:12 D and 21 ± 1 °C. High-salt treatment was performed in 120 mM sodium chloride at different time points. The cell growth rate was assessed by measuring OD_750._

### 2.2. High-Performance Liquid Chromatography-Tandem Mass Spectrometry (HPLC-MS/MS) Analysis

The frozen *U. prolifera* samples were pulverized in liquid nitrogen. About 0.5 g of the powder was aliquoted into a tube and 5 mL of acetone was added. After ultrasonication in a water bath for 15 min, the sample was centrifuged. The upper phase was collected, and the precipitate was repeatedly extracted until it was colorless. The resulting sample solution was dried with nitrogen gas. The dry extracts were dissolved in 25 mL of methanol and then filtered through a 0.22 μm polyvinylidene fluoride (PVDF) membrane filter. All the samples were analyzed using HPLC-MS/MS.

We employed an Agilent 1290 high performance liquid chromatograph in tandem (Agilent, Santa Clara, CA, USA) with an AB Qtrap6500 mass spectrometer (Allen-Bradley, Milwaukee, WI, USA)to quantify lutein, zeaxanthin, violaxanthin and antheraxanthin contents. The samples were injected into a CORTECS UPLC C18+ column (1.6 μm, 2.1 mm × 75 mm) and maintained at 35 °C. Mobile phase A consisted of 0.1% (*v*/*v*) formic acid in water and mobile phase B consisted of methanol. The column was eluted at a flow rate of 450 µL/min, with the gradient elution profile starting at 10% A and 90% B for 3 min. The proportion of mobile phase B was then increased to 100%, and mobile phase A was decreased to 0%. Finally, the initial mobile phase proportions were restored within 4 min to equilibrate the column for a further 2 min. The samples were then analyzed using atmospheric pressure chemical ionization (APCI) in the positive mode under the following conditions: ion source temperature TEM (transmission electron microscopy), 380 °C; curtain gas, 40 Psi; nebulizer gas (Gas1), 50 Psi; auxiliary heating gas (Gas2), 55 Psi; and collision gas CAD (collision activated dissociation), medium. The parameters for the MS analysis of xanthophylls are listed in Appendix A. The relative contents of xanthophylls were quantified by integrating and comparing peak areas with those of the matrix-matched calibration curve.

### 2.3. RNA Isolation and qRT-PCR

The total RNA was extracted from *U. prolifera* and *C. reinhardtii* using TRIzol reagents (Thermo Fisher, Waltham, MA, USA), followed by reverse-transcription into cDNA using the Hifair™ II 1st Strand cDNA Synthesis Kit (Yeasen, Shanghai, China). Then, the levels of transcripts were detected using the Hieff^®^ qPCR SYBR Green Master Mix (Low Rox Plus) (Yeasen, Shanghai, China). The primers used for qRT-PCR are listed in Appendix A. The *18Sr RNA* and *Tubulin* were used as the internal reference genes for *U. prolifera* and *C. reinhardtii*, respectively. All the samples were analyzed in triplicate, and the 2^−ΔΔCT^ method was used to determine the transcript levels.

### 2.4. Full-Length of the UpZEP Gene Cloning and Bioinformatics Analysis

The sequence of the *ZEP* gene was obtained from the *U. prolifera* genome [31]. The full-length sequences and coding DNA sequences (CDS) of the *UpZEP* gene were amplified using the genomic DNA and cDNA from *U. prolifera* as templates, respectively. The primers were designed using Primer Premier 5.0 and are listed in Appendix A. The physicochemical properties of the *UpZEP* protein were analyzed using the Prot Param program [32]. To analyze the phylogenetic relationship, *ZEP* protein sequences from representative Plants, Chlorophyta, Bacillariophyta, Rhodophyta and Cyanobacteria were downloaded from NCBI. Multiple sequence alignment was performed using ClustalW2.1, and the phylogenetic tree was constructed using the neighbors-joining method with MEGA6 (Bootstrap value = 1000). The Newick tree was exported from MEGA6 and improved using the Evolview website (https://evolgenius.info//evolview-v2 (accessed on 26 July 2024)).

### 2.5. C. reinhardtii Transformation and Screening of Positive Clones

We used the pChlamy-3 vector, an expression vector containing the Hsp70A-Rbc S2 fusion promoter, to explore the function of the *UpZEP* gene. The FAD (flavin adenine dinucleotide) domain coding sequences of the *UpZEP* gene were amplified to construct the recombinant vector pChlamy-3_*UpZEP*. Then, the pChlamy-3_*UpZEP* recombinant expression vector was transformed into *C. reinhardtii* using the “glass bead transformation method”, and monoclonal algal colonies were selected on hygromycin-B-containing medium (50 mg mL^−1^). The hygromycin-B-resistant and wild-type strains of *C. reinhardtii* were randomly selected, genomic DNA was extracted and the *UpZEP* gene sequences were amplified using gene-specific primers (Appendix A).

### 2.6. Functional Characterization via Heterologous Expressions of the UpZEP Gene in Yeasts

The FAD domain coding sequences of the *UpZEP* gene were recombined into the pYES2 vector. The recombinant plasmid pYES2-*UpZEP* was transformed into the *S. cerevisiae* strain INVSc1 (Coolaber, Beijing, China), and yeast cells harboring pYES2-*UpZEP* were selected on SD-Ura medium (Coolaber, Beijing, China). Single colonies were picked and transferred into liquid SD-Ura medium, then grown overnight at 200 rpm in a shaker at 30 °C. An aliquot of 1 mL of the suspension was centrifuged, resuspended in 100 µL of ddH_2_O and added to SG-Ura medium (Coolaber, Beijing, China) containing different concentrations of NaCl to observe growth.

### 2.7. Measurement of Chlorophyll Contents in C. reinhardtii

The samples were dissolved in 95% ethanol and crushed using an ultrasonic homogenizer. Then, they were placed in a dark environment at 4 °C for 24 h, and the absorbance values at OD665 and OD649 were measured. The contents of chlorophyll were calculated using the following equations:
Chl. *a* = 13.95 × OD665 − 6.88 × OD649Chl. *b* = 24.96 × OD649 − 7.32 × OD665Total chlorophyll = Chl. *a* + Chl. *b*

### 2.8. Statistical Analyses

All statistical analyses were conducted using Tukey’s test with GraphPad Prism software. Statistical significance was indicated by *p*-values of *p* < 0.05, *p* < 0.01, *p* < 0.001 and *p* < 0.0001, marked as *, **, *** and ****, respectively. Three biological replicates were performed for all experimental treatments. The results are expressed as the mean ± standard deviation (SD).

## 3. Results

### 3.1. Determination of Relative Content of Xanthophylls in U. prolifera

The relative contents of xanthophylls in *U. prolifera* grown under different salinity conditions were tested using HPLC-MS/MS technology. The results indicated the presence of several kinds of xanthophylls in *U. prolifera*, such as lutein (L), zeaxanthin (Z), antheraxanthin (A) and violaxanthin (V), with lutein and zeaxanthin being the main components. Compared to the control group (25‰), the relative contents of lutein, zeaxanthin and violaxanthin increased significantly (*p* < 0.05) under high-salt stress (50‰), whereas the content of antheraxanthin remained unchanged (Figure 1A–D). Under high-salt stress (50‰), the content of de-epoxidized xanthophylls increased significantly (*p* < 0.05) compared to the control group (25‰), resulting in the xanthophyll cycle being in a de-epoxidation state ([Z/(V + A + Z)]; [(Z + 0.5A)/(V + A + Z)]) (Figure 1E,F).

### 3.2. Full-Length Cloning and Characterization of the UpZEP Gene

We amplified the full-length sequence and CDS of the *U. prolifera* gene using PCR (Appendix A). After a BLAST search, the gene was identified as the *ZEP* gene and named *UpZEP*. The full-length sequence of the *UpZEP* gene is 1609 bp, containing three exons and two introns, with a CDS of 1122 bp, encoding 373 amino acids. The ProtParam tool was used to analyze the properties of the protein, revealing that the molecular mass of the *UpZEP* protein is about 41.37 kDa, and the theoretical isoelectric point is 5.68.

The *UpZEP* protein sequences were aligned with *ZEP* proteins from *A. thaliana*, *Oryza sativa*, *Dunaliella tertiolecta, Haematococcus lacustris* and *C. reinhardtii*. As shown in Figure 2, the amino acid sequence of *UpZEP* exhibits high homology with their *ZEP*s. Furthermore, there are two highly conserved regions, which were predicted to be the FAD domain and the FHA (fork head-associated) domain, respectively. This suggests that *UpZEP* likely performs the same role as other *ZEP*s, catalyzing epoxidation using NADPH and FAD as cofactors [33].

In order to further investigate the evolutionary relationship of *UpZEP*, MEGA 6 software was used to construct a phylogenetic tree of *ZEP*s from several Plants, Chlorophyta, Rhodophyta, Bacillariophyta and Cyanobacteria using the neighbor-joining method. The results showed that *UpZEP* clustered with other Chlorophyta *ZEP*s in a single branch, with *UpZEP* being most closely related to the *ZEP* of *D. tertiolecta* (Figure 3). *UpZEP* and other Chlorophyta *ZEP*s were evolutionarily closer to the *ZEP*s of land plants. These results suggest that the function of *UpZEP* may be similar to the *ZEP*s of Chlorophyta and land plants.

### 3.3. Expression Levels of the UpZEP Gene under High-Salt Stress

In order to investigate the expression patterns of the *UpZEP* gene under high-salt stress (50‰), the expression of the *UpZEP* gene was analyzed at 12 h, 24 h and 48 h under different salinity conditions. The internal reference gene used was *18Sr RNA*. Compared to the control group (25‰), high-salt stress (50‰) significantly induced the expression of the *UpZEP* gene (*p* < 0.01). The gene expression reached maximum at 24 h, then decreased at 48 h but remained significantly higher than in the control group (25‰) (*p* < 0.01) (Figure 4).

### 3.4. Functional Characterization via Heterologous Expressions of the UpZEP Gene in Yeast

To investigate whether heterologous overexpression of the *UpZEP* gene affects high-salt tolerance in yeast, the recombinant vector pYES2-*UpZEP* was transformed into *S. cerevisiae* INVSc1 strains. Yeasts transformed with pYES2-*UpZEP* (experimental group) and those with empty pYES2 (control group) were respectively cultured on SG-Ura medium with different salinities. The results showed that both strains could grow on SG-Ura medium containing 0 M, 0.5 M and 1 M NaCl. On the SG-Ura medium with 0 M NaCl, there was no difference in the biomass between the two groups. On the SG-Ura medium with 0.5 M and 1 M NaCl, the biomass of the strains in the experimental group increased significantly compared to the control group; however, on the SG-Ura medium with 1.3 M NaCl, the control strains showed no growth, whereas the experimental strains were able to grow (Figure 5).

### 3.5. Validation of C. reinhardtii Transformation and UpZEP Gene Expression Levels under High-Salt Stress

To further explore the biological function of the *UpZEP* gene in vivo, we generated a transgenic *C. reinhardtii* strain with heterologous overexpression of the *UpZEP* gene. Then, the phenotype, physiological parameters and gene expression patterns of transgenic *C. reinhardtii* were analyzed and compared to the wild-type. Firstly, the overexpression vector pChlamy-3-*UpZEP* was constructed and transformed into *C. reinhardtii* using the “glass bead transformation method”. Then, hygromycin-B-resistant monoclonal algal colonies were obtained and screened using PCR amplification with special primers (Appendix A). The selected monoclonal colonies produced target bands of the expected size, whereas the wild-type colonies did not produce bands, suggesting that *UpZEP* was successfully integrated into the genome of *C. reinhardtii* (Figure 6A).

To explore the expression patterns of the *UpZEP* gene in the transgenic *C. reinhardtii* strain, algae cultured under normal conditions were used as the control group, and strains cultured under 120 mM NaCl conditions were used as the experimental group. The expression levels of the *UpZEP* gene at 24 h, 48 h and 72 h in the two groups were detected using a qRT-PCR assay. *Tubulin* was used as the internal reference gene. The results showed that the transgenic *C. reinhardtii* exhibited a short adaptation period under high-salt stress. At 24 h, there was no significant difference in *UpZEP* gene expression between the two groups. However, compared to the control group, the *UpZEP* transcription level in the experimental group was significantly increased (*p* < 0.01) and showed an upward trend at 48 h and 72 h, indicating that the *UpZEP* gene was induced by high-salt stress (Figure 6B).

### 3.6. Analysis of High-Salt Stress Tolerance in Transgenic C. reinhardtii

In order to explore the high-salt stress resistance of *C. reinhardtii* containing the *UpZEP* gene, wild-type and transgenic *C. reinhardtii* strains were cultured under 120 mM NaCl for 6 d. The color of transgenic *C. reinhardtii* changed from light green to dark green, whereas the wild-type did not change color. The transgenic *C. reinhardtii* appeared greener than the wild-type (Figure 7A). Additionally, biomass was measured using OD_750_ light absorption values (Figure 7B). The results showed that the biomass of transgenic *C. reinhardtii* was higher than that of the wild-type after 1 d of exposure to 120 mM NaCl. The corresponding phenotypic observations and biomass measurements were consistent, indicating that the *UpZEP* gene enhanced the salt tolerance of *C. reinhardtii*.

Simultaneously, the wild-type and transgenic *C. reinhardtii* strains were cultured under 120 mM NaCl for 0, 1, 2 and 3 d, and the chlorophyll content was measured. The transgenic *C. reinhardtii* had a short adaptation period under high-salt stress. At 1 d, there was no significant difference in chlorophyll content between the transgenic and wild-type strains; however, the chlorophyll content of transgenic *C. reinhardtii* was significantly higher compared to the wild-type strains at 2 and 3 d (Figure 7C).

## 4. Discussion

Under abiotic stress, the decreased growth rate of plants and algae is often associated with reduced photosynthesis [34]. This decline in photosynthesis in plants and algae will lead to an increase in excess light energy. If the excess excitation energy is not dissipated in time, this will lead to the accumulation of ROS, resulting in oxidative damage [16]. To avoid or minimize this photoinhibition damage, plants have evolved a variety of protective mechanisms, with carotenoids playing an important role in plant photoprotection [17]. Carotenoids dissipate excess light energy through NPQ, which is mediated by zeaxanthin [13,17]. In plants and algae, carotenoids are synthesized and accumulated in the thylakoid membrane, especially within the LHC [18]. Carotenoids are classified into carotenes and xanthophylls, and *α*-and *β*-carotene can be converted into xanthophylls. Seaweeds, a special class of marine plants, each have their own unique carotenoid composition [35]. In this study, we analyzed the carotenoids of *U. prolifera*, a dominant green tide species in China. The results showed that the carotenoids of *U. prolifera* are mainly lutein and zeaxanthin, and salt stress induced an increase in the contents of lutein, zeaxanthin and violaxanthin. In our previous study, we also found that low-light and high-salt stress promoted carotenoid synthesis [36]. In addition, carotenoid synthesis is induced by copper stress in *Ulva compressa* [37]. Collectively, these findings suggest that abiotic stress commonly promotes the synthesis and accumulation of carotenoids in *Ulva*. Under abiotic stresses, such as light stress, salinity stress and nitrogen deprivation, the carotenoid content also increased in *D. salina*, which is thought to be a strategy for resisting abiotic stress [38]. In this study, the carotenoid content increased under high-salt stress, indicating that their important role in the salt tolerance of *U. prolifera*.

Lutein is considered to be a xanthophyll involved in NPQ and acts as a direct quencher of chlorophyll *a* [38]. However, zeaxanthin is an allosteric modulator of non-photochemical energy dissipation [39]. In *U. pertusa,* accumulated zeaxanthin protects the thylakoid membrane and enhances thermal quenching during desiccation [40]. Slow zeaxanthin accumulation in *U. prolifera* under high-light stress is considered atypical for NPQ [41]. In this study, high-salt stress significantly increased the relative content of xanthophylls, leading to the accumulation of zeaxanthin, which may have a positive effect on the high-salt stress response of *U. prolifera*. The xanthophyll cycle significantly contributes to photoprotection in plants [19]. Violaxanthin and antheraxanthin, the epoxidation products of zeaxanthin, are the key precursors of carotenoids and also participate in the xanthophyll cycle. The catalytic function of *ZEP* has diversified through plant evolution [42]. The *ZEP* gene exists widely in photosynthetic eukaryotes. At present, the *ZEP* gene has been identified in several plants and algae. The genome of the diatom *Phaeodactylum tricornutum* harbors three *ZEP* genes, but only one *ZEP* gene is involved in NPQ [43]. *Nannochloropsis oceanica* contains two *ZEP* genes [44], and one *ZEP* gene has been identified in *D. tertiolecta* [33]. In this study, the *UpZEP* gene was cloned from *U. prolifera*. The amino acid sequence of *UpZEP* exhibited high homology with other *ZEP*s, and the *UpZEP* protein contained FAD and FHA domains, indicating that *UpZEP* is a typical *ZEP* protein. Phylogenetic analysis showed that the *UpZEP* clustered into a clade with *ZEP*s from green algae, and most closed to *D. tertiolecta*. Although red algae has no xanthophyll cycle, it has a *ZEP* gene that is not the same as that in terrestrial plants and other algae. This *ZEP* introduces an epoxide group into zeaxanthin and produces antheraxanthin instead of violaxanthin, indicating that the *ZEP* gene of red algae is relatively primitive. At the same time, this also indicates that *ZEP* first only catalyzed the transformation of zeaxanthin into antheraxanthin during the evolutionary process, and later acquired the ability to catalyze antheraxanthin into violaxanthin [42].

The *ZEP* gene is involved in the synthesis of carotenoids and ABA and plays a vital role in plant responses to abiotic stress [22,23]. In *D. tertiolecta*, with increasing salinity, the expression of the *ZEP* gene increased gradually [45]. In this study, *UpZEP* gene expression was up-regulated under high-salt stress, which is consistent with that of *D. tertiolecta* under salt stress, indicating that *UpZEP* gene expression was induced by salt stress in *U. prolifera*. However, down-regulation of the *ZEP* gene is thought to be associated with chilling sensitivities in rice [46]. Hoang et al. [47] proposed that the reversible down-regulation of *ZEP* gene expression plays an indispensable role in plants, and this reversible down-regulation mechanism is a common mechanism in plants under various light conditions at room temperature. Considering the existence of these opposite expression patterns, the role of the *UpZEP* gene *U. prolifera* under salt stress warrants further in-depth research.

Cao et al. reported that overexpression of the *ZEP* gene from *Medicago sativa* enhanced the tolerance of transgenic tobacco to low-light stress [28]. Overexpression of the alfalfa *ZEP* gene conferred drought and salt tolerance in transgenic tobacco [29]. Park et al. [48] found that overexpression of the *AtZEP* gene made transgenic *A. thaliana* plants grow more vigorously under salt and drought stress. In this study, the *UpZEP* gene was transformed into yeast cells and was overexpressed. The results showed that *UpZEP* transgenic yeast exhibited growth on SG-Ura medium containing 1.3 M NaCl, whereas the empty pYES2 vector transgenic yeast strain did not, indicating that overexpression of the *UpZEP* gene enhances salt tolerance in yeast. Meanwhile, the *UpZEP* gene was heterogeneously overexpressed in *C. reinhardtii*. Transcription level analysis showed that the *UpZEP* gene expression level was significantly up-regulated under high-salt stress, indicating that the gene was induced by salt stress. Under continuous salt stress, transgenic *C. reinhardtii* had a higher survival rate and chlorophyll content than the wild-type. These results suggest that the *UpZEP* gene enhances plant salt tolerance. As a euryhaline marine organism, the discovery of the *Ulva* salt-tolerant gene will be useful for breeding salt-tolerant crops in the future. For example, functional expression of an animal type-Na+-ATPase gene from the red seaweed *Porphyra yezoensis* increases salinity tolerance in rice plants [49].

## 5. Conclusions

The current study analyzed the relative content of xanthophylls in *U. prolifera* under different salinity conditions. We found that high-salt stress significantly increased the relative content of xanthophylls. A *ZEP* gene was cloned and identified from *U. prolifera*, and the *UpZEP* gene expression was significantly induced by high-salt stress. Moreover, heterologous overexpression of the *UpZEP* gene in yeast and *C. reinhardtii* increased their tolerance to salt. These results provide a new perspective for studying the resistance mechanisms of *U. prolifera* to high-salt stress.

## Figures and Tables

**Figure 1 biology-13-00695-f001:**
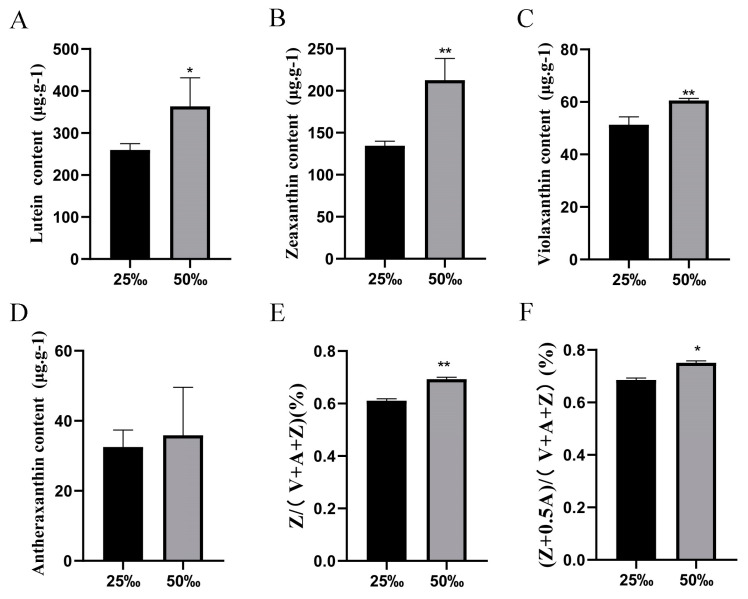
Relative content of xanthophylls in *U. prolifera*. (**A**) Lutein content; (**B**) Zeaxanthin content; (**C**) Violaxanthin content; (**D**) Antheraxanthin content; (**E**) De-epoxidation state, Z/(V + A + Z); (**F**) De-epoxidation state, (Z + 0.5A)/(V + A + Z). Zeaxanthin, Z; Violaxanthin, V; Antheraxanthin, A. 25‰, control group; 50‰, high-salt stress. Values are means ± SD (n = 3; compared with 25‰, * *p* < 0.05, ** *p* < 0.01, Student’s *t*-test). All experiments were repeated at least three times with similar results.

**Figure 2 biology-13-00695-f002:**
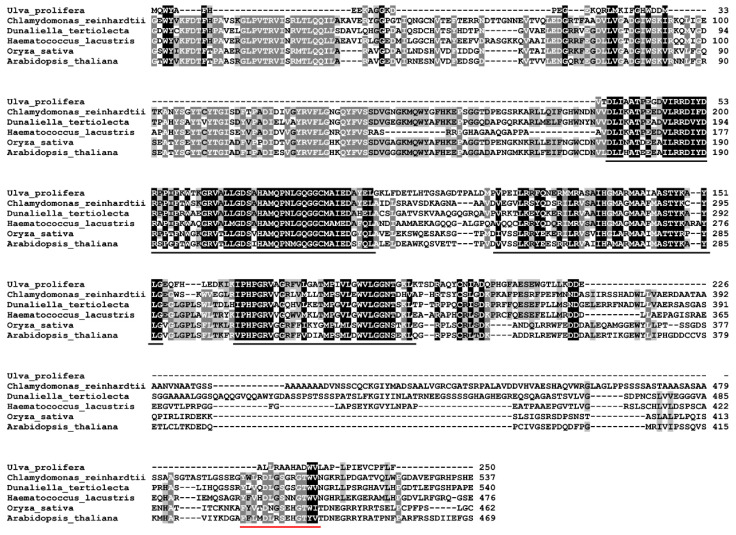
Alignment of the amino acid sequence of *UpZEP* with other *ZEP*s. The residues that are identical in more than 80% of all the sequences are shaded in gray. The conserved residues are shaded in black. The FAD domain is indicated with a black underline, and the FHA domain is indicated with a red underline. The species and corresponding GenBank accession number are as follows: *Chlamydomonas reinhardtii* (AAO48940.1); *Dunaliella tertiolecta* (AZK89898.2); *Haematococcus lacustri*s (GFH20616.1); *Oryza sativa* (BAB39765.1); *Arabidopsis thaliana* (AAG38877.1).

**Figure 3 biology-13-00695-f003:**
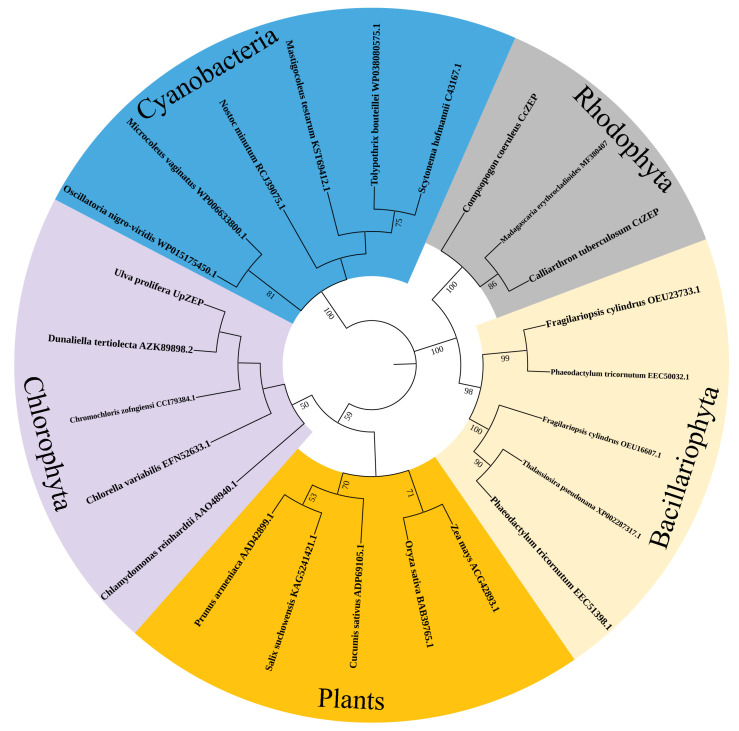
Phylogenetic trees of *ZEP*s from several Plants, Chlorophyta, Rhodophyta, Bacillariophyta and Cyanobacteria. The neighbor-joining method was used to reconstruct the phylogenetic trees using the MEGA6 software (Bootstrap value = 1000). *UpZEP* refers to *U. prolifera*. The GenBank IDR is shown right after each species name.

**Figure 4 biology-13-00695-f004:**
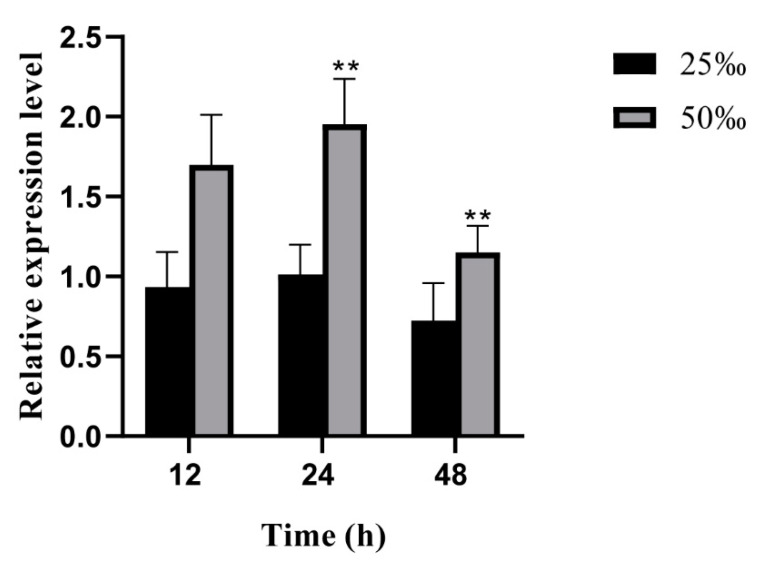
Expression levels of the *UpZEP* gene under different salinity conditions at 12 h, 24 h and 48 h. The 25‰ condition represents the control group and the 50‰ condition represents high-salt stress. The values are expressed as means ± SD (n = 3; compared with 25‰, ** *p* < 0.01, Student’s *t*-test).

**Figure 5 biology-13-00695-f005:**
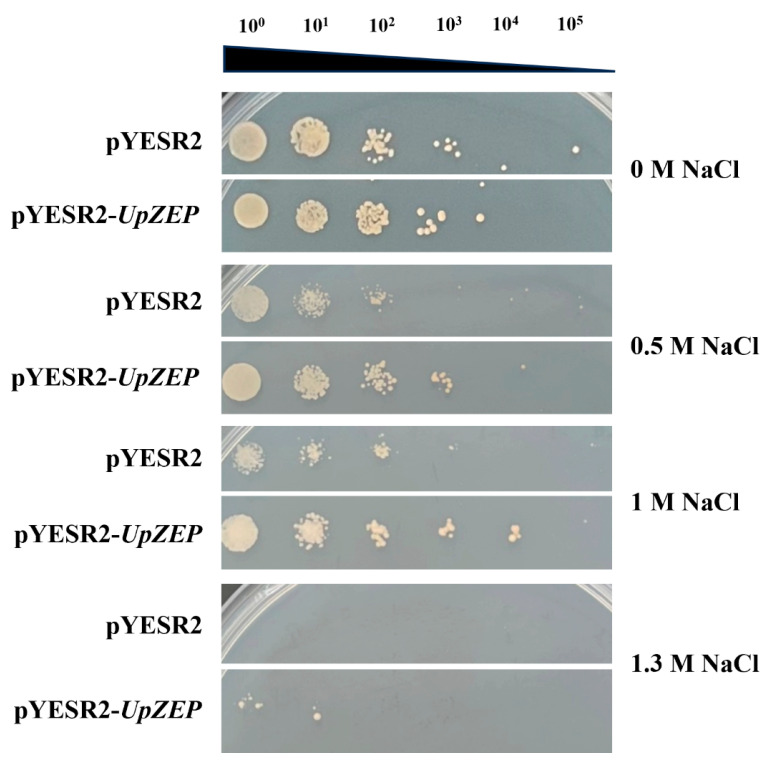
Verification experiment of yeast phenotypes. Yeast strains transformed with empty vector pYESR2 were used as the control group. Yeast strains transformed with recombinant vector pYESR2-*UpZEP* were used as the experimental group. Transformants of pYESR2 and pYESR2-*UpZEP* were grown in SG-Ura medium supplemented with different concentrations of NaCl.

**Figure 6 biology-13-00695-f006:**
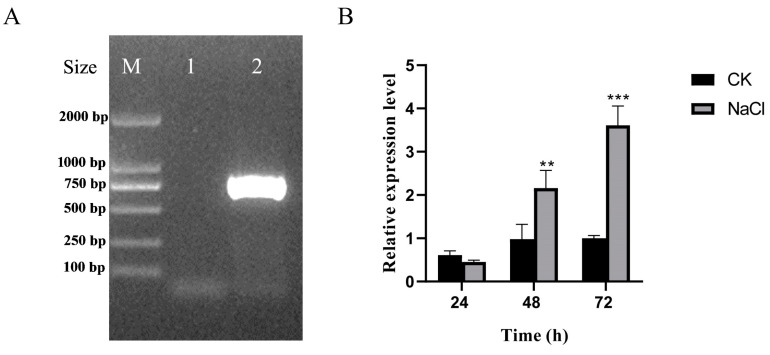
Validation of transgenic *C. reinhardtii* and the expression pattern of the *UpZEP* gene under high-salt stress. (**A**) Validation of transgenic *UpZEP*-positive clones using gel electrophoresis: 1, wild-type *C. reinhardtii*; 2, transgenic *UpZEP*-positive *C. reinhardtii*. (**B**) Expression levels of the *UpZEP* gene under high-salt stress. CK, control group; NaCl, 120 mM. The values are expressed as means ± SD (n = 3; compared with CK, ** *p* < 0.01, *** *p* < 0.001, Student’s *t*-test).

**Figure 7 biology-13-00695-f007:**
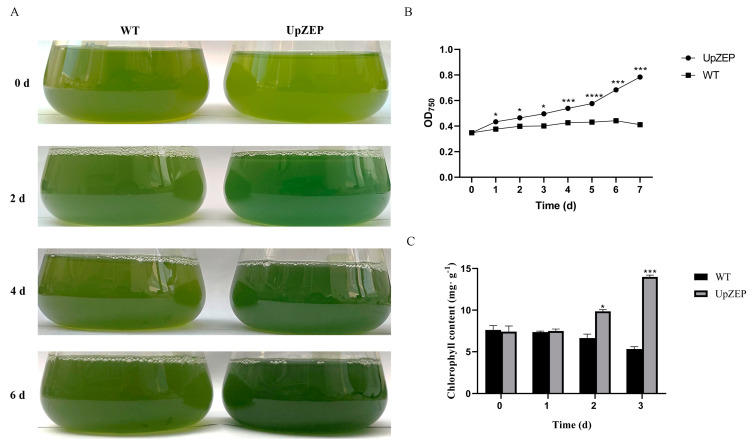
The biomass and chlorophyll content in *C. reinhardtii* under high-salt stress. Transgenic and wild-type (WT) *C. reinhardtii* phenotypes (**A**), biomass measurements (**B**) and chlorophyll content (**C**). WT, control group. The values are expressed as means ± SD (n = 3; compared with WT, * *p* < 0.05, *** *p* < 0.001, **** *p* < 0.0001, Student’s *t*-test).

## Data Availability

The datasets generated and/or analyzed during this study are available from the corresponding author on reasonable request.

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
