# Peer review of "Cloning and Functional Analysis of a Zeaxanthin Epoxidase Gene in Ulva prolifera"

_biology, 2024, doi:10.3390/biology13090695_

Round 1

Reviewer 1 Report

Comments and Suggestions for Authors

See file.

Comments on the Quality of English Language

Please, improve your language.

Author Response

Comments 1: “and” ?

Response 1: Thank you for pointing this out. We have deleted “and” in the revised manuscript. Please check the revised manuscript-line 5.

Comments 2: “…which changes the conformation of LHC…” its conformation

It would be better to separate sentence into two.

Response 2: Thank you for your reminder. We have revised in the revised manuscript. Please check the revised manuscript-line 75-77.

Comments 3: “Pure lines were obtained by genetic and morphological identification, and cultured in our laboratory [31]”

Reference [31] contains no info about cultivation, but only natural samples. It must be deleted. Ref. [32] is enough.

Response 3: Thank you for your reminder. We agree with this comment. We have deleted “reference [31]” in the revised manuscript. Please check the revised manuscript-line 109.

Comments 4: Authors use different salinity units (‰, M and mM) for Ulva, Chlamydomonas and Saccharomyces respectively. Salt concentration in the last two spp. differ by 10 times.

Response 4: Thank you for pointing this out. We determine the salt concentration by pre-testing. Chlamydomonas and Saccharomyces may have different tolerance to salinity.

Comments 5: Do not understand why authors include abbreviations – TEM and CAD, and never mention them later.

Response 5: Thank you for pointing this out. We have revised in the revised manuscript. Please check the revised manuscript-line 140 and 142.

Comments 6: “The sequence of the ZEP gene was obtained from the U. prolifera genome [33].” Paper [33] do not contain the information on ZEP gene.

Response 6: Thank you for pointing this out. In this paper, the ZEP gene sequence was obtained by analyzing the genome sequencing data of U. prolifera in paper [33].

Comments 7: No interpretation of the abbreviation FAD. It appears for the first time only at the line 219.

Response 7: Thank you for pointing this out. We have revised in the revised manuscript. Please check the revised manuscript-line 168 and 225.

Comments 8: “The contents of chlorophyll were measured according to the methods of Yang [35]”

Calculations of measured chlorophyll contents in the presented paper do not coincide with the same in reference [35]. Why?

Response 8: Thank you for pointing this out. We have revised in the revised manuscript. Please check the revised manuscript-line 185-188.

Comments 9: “the amino acid sequence of UpZEP has high homology with other ZEPs”

The high homology is seen for Dunaliella and Chlamydomonas. Why do you think so for Ulva?

Response 9: Thank you for pointing this out. UpZEP protein sequences were aligned with ZEP proteins from Dunaliella and C. reinhardtii. The results shown that the amino acid sequence of UpZEP has high homology with their ZEPs. UpZEP gene was cloned from U.prolifera genome.

Comments 10: “Tress” instead trees. Additionally, must be include “UpZEP gene” in the legend according to the text.

It is not clear why authors compare protein sequences and than demonstrate the phylogenetic tree of ZEP genes. The explanation of the details must be extended.

Response 10: Thank you for your reminder. We have revised in the revised manuscript. Please check the revised manuscript-line 244, 247, and 235-242.

Comments 11: “the UpZEP was successful integration”. Must be was successfully integrated.

Response 11: Thank you for pointing this out. We have revised in the revised manuscript. Please check the revised manuscript-line 288.

Comments 12: The descriotion of the experiment should be given in more details.

“Then” (?) Simultaneously, At the same, Beside time etc.

Response 12: Thank you for pointing this out. We have revised in the revised manuscript. Please check the revised manuscript-line 307-316.

Comments 13: “it” (?)

Response 13: Thank you for your reminder. We have revised in the revised manuscript. Please check the revised manuscript-line 371.

Reviewer 2 Report

Comments and Suggestions for Authors

The paper focuses on studying the contribution of zeaxanthin epoxidase to salinity tolerance of the green alga Ulva prolifera.  Based on their results, the authors conclude that this enzyme and the processes associated with the xanthophyll cycle play an important role in accommodation of Ulva to enhanced salinity. The data look convincing, and the paper can be recommended for publication in “Biology” after a minor revision.

The specific comments are in the attached file. Also, in the attached file I suggested some correction of English.

Comments on the Quality of English Language

Generally, English is acceptable, though some parts of the manuscript (e.g., Abstract) need polishing, as several sentences are unclear and should be revised. In the attached file I suggested correction of the most serious mistakes, but there are still some problems, so the authors are advised to carefully check all the text.

Reviewer 3 Report

Comments and Suggestions for Authors

In this manuscript, the authors cloned the Zeaxanthin epoxidase gene from Ulva prolifera and studied its impact on salt stress by heterologously overexpressing it in Chamy and yeast. They also showed the differential expression of the gene upon salt stress.

Although the work is soundly performed, it bears minimal importance and novelty. The gene has been widely studied and tested for its function in salt and other abiotic stresses. The only novelty of the study is that the authors cloned the gene from another organism. 

Here are a few comments that the authors need to address to improve the manuscript.

1) What is the rationale for cloning the Zeaxanthin epoxidase from the U. prolifera when the protein is similar to the other orthologs?

2) Please expand figures legends, figures 1,3 and 4 with more description.

3) Provide the bootstrap number to each branch of the phylogenetic tree in Figure 3.

4) In the Figure and its relevant text, please explain 25% and 50%. What is control? What gene(s) is used as a reference for the qRT-PCR? The result section must include this basic information to understand the data better.

5) In Figure 5, the overexpressed line showed better growth at 1M NaCl than 0.5 M. Can you please explain or stipulate this phenomenon? How many times was this experiment repeated?

6) For expressing in Chamy, what promoter did the authors use to overexpress the gene? Please provide a brief introduction of the promoter in the text.

7) What reference genes are used to generate qRT-PCR data for Figure 6?

8) As per Figure 7B, the control chamy cells resisted such a high salt treatment for over a week. Any explanation for this phenomenon?

9) Discussion sections feel more like introductions. The authors need to modify the discussion section to discuss the result in detail rather than providing a further introduction to the gene or xanthophyll cycle.

Comments on the Quality of English Language

The English language needs minor corrections.

Author Response

Comments 1: What is the rationale for cloning the Zeaxanthin epoxidase from the U. prolifera when the protein is similar to the other orthologs?

Response 1: Thank you for pointing this out. Based on the previous transcriptome sequencing data of our research group, we used Zeaxanthin epoxidase as the key word to obtain the gene sequence, and then compared the amino acid sequence of the obtained open reading frame with the NCBI Protein-blast data to determine the coding region of this gene. Finally, the ZEP gene was cloned by PCR.

Comments 2: Please expand figures legends, figures 1, 3 and 4 with more description.

Response 2: Thank you for pointing this out. We have revised in the revised manuscript. Please check the revised manuscript.

Comments 3: Provide the bootstrap number to each branch of the phylogenetic tree in Figure 3.

Response 3: Thank you for pointing this out. We have revised in the revised manuscript. Please check the revised manuscript.

Comments 4: In the Figure and its relevant text, please explain 25% and 50%. What is control? What gene(s) is used as a reference for the qRT-PCR? The result section must include this basic information to understand the data better.

Response 4: Thank you for pointing this out. We have revised in the revised manuscript. Please check the revised manuscript-line 203-207, 250-254, and 294.

Comments 5: In Figure 5, the overexpressed line showed better growth at 1M NaCl than 0.5 M. Can you please explain or stipulate this phenomenon? How many times was this experiment repeated?

Response 5: Thank you for pointing this out. At the same dilution concentration, the growth status was determined by comparing the overall size and density of plaque. We repeated the experiment three times.

Comments 6: For expressing in Chamy, what promoter did the authors use to overexpress the gene? Please provide a brief introduction of the promoter in the text.

Response 6: Thank you for pointing this out. We have provided a brief introduction of the promoter in the revised manuscript. Please check the revised manuscript-line 167-168.

Comments 7: What reference genes are used to generate qRT-PCR data for Figure 6?

Response 7: Thank you for pointing this out. Tubulin was used as the internal reference gene. We have supplemented in the revised manuscript. Please check the revised manuscript-line 294.

Comments 8: As per Figure 7B, the control chamy cells resisted such a high salt treatment for over a week. Any explanation for this phenomenon?

Response 8: Thank you for pointing this out. We have revised in the revised manuscript. Please check the revised manuscript-line 311-315.

Comments 9: Discussion sections feel more like introductions. The authors need to modify the discussion section to discuss the result in detail rather than providing a further introduction to the gene or xanthophyll cycle.

Response 9: Thank you for pointing this out. We have revised in the revised manuscript.

Please check the revised manuscript-line 338-339, 341-346, 370-376, and 401-404.